# Transjugular Intrahepatic Portosystemic Shunt as a Bridge to Abdominal Surgery in Cirrhosis

**DOI:** 10.3390/jcm13082213

**Published:** 2024-04-11

**Authors:** Fabio Melandro, Simona Parisse, Stefano Ginanni Corradini, Vincenzo Cardinale, Flaminia Ferri, Manuela Merli, Domenico Alvaro, Francesco Pugliese, Massimo Rossi, Gianluca Mennini, Quirino Lai

**Affiliations:** 1Department of General and Specialist Surgery, Sapienza University of Rome, 00185 Rome, Italy; f.pugliese@uniroma1.it (F.P.); massimo.rossi@uniroma1.it (M.R.); gianluca.mennini@uniroma1.it (G.M.); quirino.lai@uniroma1.it (Q.L.); 2Department of Translational and Precision Medicine, Sapienza University of Rome, 00185 Rome, Italy; simona.parisse@uniroma1.it (S.P.); stefano.corradini@uniroma1.it (S.G.C.); vincenzo.cardinale@uniroma1.it (V.C.); flaminia.ferri@uniroma1.it (F.F.); manuela.merli@uniroma1.it (M.M.); domenico.alvaro@uniroma1.it (D.A.)

**Keywords:** transjugular portosystemic shunt, abdominal surgery, portal hypertension, liver transplantation

## Abstract

Abdominal surgery is associated with high postoperative mortality and morbidity in cirrhotic patients. Despite improvements in surgical techniques, clinical management, and intensive care, the outcome could be influenced by the degree of portal hypertension, the severity of hepatopathy, or the type of surgery. Preoperative transjugular intrahepatic portosystemic shunt (TIPS) placement, in addition to medical therapy, plays an important role in managing the complications of portal hypertension such as ascites, hepatic encephalopathy, variceal bleeding or portal vein thrombosis. To date, the improvement of post-surgery outcomes in cirrhotic patients after TIPS placement remains unclear. Only observational data existing in the literature and prospective studies are urgently needed to evaluate the efficacy and safety of TIPS in this setting. This review aims to outline the role of TIPS as a tool in postoperative complications reduction in cirrhotic patients, both in the setting of emergency and elective surgery.

## 1. Introduction

Patients with cirrhosis and portal hypertension face increased risk during surgical procedures, especially major surgeries, as it can lead to acute decompensation and postoperative complications, resulting in high rates of morbidity and mortality [1].

To manage complications related to portal hypertension, transjugular intrahepatic portosystemic shunt (TIPS) became a percutaneous radiological procedure widely used in the last decades.

TIPS diverts the bloodstream directly from the portal system to the systemic circulation using an artificial channel, resulting in a lower portosystemic pressure gradient (PSPG) below conventional thresholds of <12 mmHg [2].

To date, the impact of preoperative TIPS on patients requiring elective and emergency surgery remains uncertain [3].

Surgical patients with cirrhosis present, typically, a complex clinical scenario, often characterized by multifactorial comorbidities and both splanchnic and systemic hemodynamics dysfunctions.

Abdominal surgery, both hepatic and non-hepatic surgery, is associated with relevant postoperative morbidity and mortality in cirrhotic patients. Postoperative mortality ranges between 0 and 25% in Child–Turcotte–Pugh (CTP) class A, and more than 50% in the worst classes. Postoperative morbidity was around 30% in CTP class A.

Significant risk factors for postoperative complications are emergency surgery, preoperative decompensated cirrhosis, ascites, low serum sodium levels, and renal and cardiopulmonary comorbidities. Portal hypertension with a PSPG of more than 10 mmHg impacts negatively in the postoperative outcomes, with an increased risk of bleeding and an intra- and postoperative blood transfusion requirement [4].

Reverter et al. [5] recently found that an elevated PSPG > 16 mmHg represents an independent risk factor for an unfavorable postoperative outcome in non-hepatic abdominal surgery.

Only a few studies explored the real impact of TIPS on complication rates after non-hepatic surgery.

Patel et al. [6], in a retrospective study on 20 patients, found that preoperative TIPS placement does not improve survival, particularly in patients with decompensated cirrhosis. Despite this, TIPS may improve liver function in patients with higher CTP scores, in order to allow surgical procedures.

On the contrary, in a recent work by Piecha et al. [7], it has been shown that TIPS is associated with a reduced incidence of postoperative in-house mortality and in-house liver transplantation needed, only in selected patients with CTP class B and C cirrhosis. The authors presented a retrospective cohort of patients who underwent surgery within 3 months after TIPS placement. The benefit of TIPS was demonstrated both in low-risk surgical procedures (hernia repair, laparoscopic surgery as cholecystectomy, abdominal exploration, hematoma evacuation, sigmoidostomy) and high-risk procedures (open abdomen surgery as gastrectomy, esophagectomy).

In patients awaiting abdominal surgery, TIPS placement can be an option to reduce postoperative complications; a careful multidisciplinary approach is mandatory in order to plan the timing of TIPS positioning and eligibility for surgery [8].

The question of whether TIPS placement improves post-surgery outcomes in abdominal procedures is still debated in the literature and deserves considerable attention.

This review aims to explore the evidence of using TIPS as a bridge to elective and emergency abdominal surgery in cirrhotic patients, highlighting different types of abdominal surgery (Table 1).

### Indication for TIPS

As it is already known, TIPS is an artificial channel, radiologically placed, connecting the portal vein to the inferior cava through the liver, in order to reduce PSGP [25,26] (Figure 1).

Today, the most frequent indications for the placement of TIPS are as follows:Refractory ascites and/or refractory hydrothorax;Treatment of gastroesophageal varices as secondary prophylaxis of variceal bleeding or as rescue therapy for uncontrolled bleeding;Bridge in patients awaiting liver transplantation, particularly in patients with portal hypertension complications. No significative difference was reported in the recent literature on the post-transplant outcome, comparing patients with TIPS and no TIPS, regarding postoperative complications, transfusion requirement, length of stay, and re-transplantation rate [27].

Other, less frequent indications are as follows:Chylous ascites and chylothorax;Gastric varices;Abdominal ectopic varices.

More recent indications are as follows:Portal vein thrombosis in patients who do not respond to anti-coagulant therapy and as an attempt in patients with an extension of the thrombosis that contraindicates liver transplantation;Recurrent ascites;Vascular disorders such as Porto-sinusoidal Vascular Disorder (PSVD) and Budd–Chiari Syndrome;Refractory ascites in liver transplant recipients. This complication occurred in 5–7% of patients. The use of post-liver transplant TIPS was reported in a large series by Saad et al. [28], including 39 cases of refractory ascites and variceal bleeding after transplant. In this series, the PSPG of 10 mmHg after TIPS placement. Bianco et al. [29] presented three cases of refractory ascites after transplant in patients without chronic disease recurrence. In all cases, refractory ascites were resolved after TIPS placement.

Complications can arise during or after the procedure [30]. Some of the potential complications include hepatic encephalopathy, occurring in approximately 35–50% of cases [31]; hemorrhage, where bleeding can occur during the insertion of the shunt or afterward; and stent dysfunction, where stent occlusion or stenosis can occur, leading to recurrent symptoms of portal hypertension. Regular surveillance with imaging is necessary to detect stent dysfunction; infections such as bacteremia or shunt-related abscesses, particularly in patients with pre-existing ascites or immunocompromised states; heart failure due to increases in venous return; pulmonary complications, as, rarely, TIPS placement can lead to pulmonary embolism or pleural effusion; and biliary complications, as in some cases, TIPS placement can lead to injury to the bile ducts, resulting in bile leaks or obstructions.

## 2. Colorectal Surgery

Colorectal surgery in cirrhotic patients is burdened by a high risk of major complications and mortality [32].

The studies available provided conflicting evidence regarding the advantages of TIPS placement in colorectal surgery.

Few large studies have addressed specific outcomes in this clinical scenario. Vinet et al. [9], in a retrospective, comparative study that included 10 patients who underwent colorectal surgery after TIPS placement (mean age 58 ± 14 years) and 13 patients without preoperative TIPS, found no benefit of preoperative TIPS placement to prevent postoperative complications and mortality. In this cohort, height patients underwent surgery due to colon cancer; the stage of liver disease was low or moderate.

Similar results were obtained in a French study [33], based on a cohort of eight cases of cirrhotic patients with severe portal hypertension, resulting in a high rate of overall complications (75%), mortality (25%), and liver decompensation (50%) after surgical procedures including right, left colectomy or proctectomy for colorectal cancer. TIPS placement was carried out between one and nine weeks before surgery. One patient experienced cancer recurrence and died 12 months after surgery.

A recent, multicenter series [4] compared 34 cirrhotic patients with TIPS and 38 patients without TIPS in surgical elective procedures. Colorectal surgery was performed in 65% of CTP class A patients, with a median age of 60 years. The TIPS group showed better 90-day postoperative mortality, intensive care unit (ICU) stay, major complications, and sepsis episodes in patients with mild hepatopathy (MELD-Na < 15), whereas in severe hepatopathy, the reduction of complications and mortality was not statistically significative. Main postoperative complications consisted of anastomotic fistula (three TIPS patients vs. seven no TIPS patients, *p* = 0.494), incisional hernia (three TIPS patients vs. one no TIPS patient, *p* = 0.314), and incisional abscess (three TIPS patients and four no TIPS patients, *p* = 1.000). The patients with less severe portal hypertension (PSPG ≤ 13 mmHg) and less advanced liver dysfunction (MELD-Na score ≤ 15) showed better results in terms of post-surgical complications when TIPS was placed, despite the absence of statistical significance.

Moreover, according to a multicenter British study published in 2021 [10], patients with higher post-TIPS PSPG underwent colon resection and showed higher risk of adverse events, such as encephalopathy, sepsis, and bleeding with 1-, 6-, and 12-month survival of 90%, 80%, and 76%, respectively.

Other small studies have investigated the role of preoperative TIPS in this scenario.

Kapeleris et al. [11] described two cases of patients planned for colorectal surgery. One case successfully underwent lower anterior resection due to adenocarcinoma. The second one, a planned total colectomy due to ulcerative colitis, was complicated by liver failure after TIPS placement, and immediately put on the liver and bowel transplantation waiting list. Interestingly, the last patient showed inadequate porto-systemic pressure reduction (from 22 to 19 mmHg) and stent thrombosis.

Related to Paul Brousse experience, [12] reported two cases of left colon resection for cancer in patients with grade 2 esophageal varices. The patients underwent surgery at least one month after TIPS placement. One patient died in the first 90 days of hepatic failure. This patient had a high CTP score before and after tips placement (before 12 and after 11). One more patient underwent post-Hartman recanalization, suffered from an intraoperative, massive hemorrhage that occurred during adesiolisis, and needed transfusions. The patient was discharged at postoperative 10 days without other complications.

Schlenker et al. [13] described a case of sigmoid colectomy for colorectal adenocarcinoma, after 4 days from TIPS placement, in a 60-year-old man with previous esophageal variceal bleeding and ascites. The postoperative course was characterized by ascites and wound infection, treated with drainage and antibiotics. Nevertheless, after 34 months of follow-up, the patient was in good condition, with a stable decrease in portosystemic gradient and intermittent encephalopathy.

A further favorable outcome was reported by Gil et al. [14] in a case of right hemicolectomy for cancer for a 63-year-old cirrhotic patient (CTP score B7), complicated by two previous bleeding episodes. The procedure was performed 30 days after TIPS placement and the results were uneventful.

Cases of mini-invasive surgery were described in this scenario; Masood et al. [15] reported a case of a 65-year-old male with a history of hepatitis C virus cirrhosis with a MELD score of 12 and a large caput-medusae secondary to portal hypertension. Two months after TIPS placement, the CT scan showed complete resolution of the periumbilical varices. The patient successfully underwent laparoscopic right hemicolectomy due to adenocarcinoma, without complications in the postoperative course.

In conclusion, among available studies, the TIPS procedure plays a role in the reduction of portal hypertension complications and has become a feasible surgical procedure, but it seems that there is no significant impact on the post-surgical outcome.

## 3. Upper-Gastrointestinal Surgery

Several experiences in cirrhotic patients who underwent upper-gastrointestinal (GI) surgery are reported in the literature, but few data are available in the TIPS placement context.

Only one large case series presented by Schmitz et al. [17] including six sleeve gastrectomies, one gastrectomy, and one esophagectomy is reported. All patients had manifestations of portal hypertension prior to TIPS (varices, ascites, or both). The mean age was 56.4 ± 8.8. Patients were managed for TIPS placement in order to improve surgical feasibility, achieving decompression of varices or reduction of ascites. After a median follow-up time of 705 days, more than 50% of patients who underwent TIPS creation underwent planned abdominal operation. The mean time between TIPS and operation was 38.7 days (range = 0–156 days). No deaths within postoperative 30 days were reported and no major surgical complications occurred.

Further publications in the form of single clinical reports or case series with a small number of patients are reported in this context: Azulay et al. [12] described one case of a patient with lower esophageal cancer, who successfully underwent esophagectomy. The patient was affected by severe portal hypertension and esophageal varices. TIPS placement allowed to perform the surgical procedure without postoperative complications. In this series, another cirrhotic patient underwent laparotomy after TIPS placement for planned cardia resection. Unfortunately, the procedure was not performed due to carcinosis, and the patients died after a few months.

Schlenker et al. [13] reported a case of subtotal distal gastrectomy and segmental transverse colectomy for gastric cancer involving the colon in 48-year-old patients affected by HCV-related cirrhosis and esophageal varices. The postoperative course was complicated by a subphrenic fecal fistula and localized peritonitis that was successfully treated with drainage and a diverting ileostomy.

A case of simultaneous gastric and sigmoid resection for synchronous gastric and colon cancer was described by Gil et al. [14] in cirrhotic 70-year-old men with a CTP score of A5. The gastro-jejunal anastomosis bleed massively during the procedure but the patient was discharged after 17 days in good condition.

No complications after surgery were also reported in a case of laparoscopic Heller miotomia for achalasia, cholecistectomia, and fonduplicatio sec. Dor, presented by de Andres et al. [18] and in a case of antrectomy for gastric antral vascular ectasia (GAVE), as reported by Becq et al. [19].

## 4. Wall Surgery

The patients with cirrhosis present a high incidence of abdominal wall hernia. The incidence rises in patients with ascites. An umbilical hernia was the most frequent condition, and 60% of patients experience recurrence following repair.

Two recent studies found a stronger benefit for patients with preoperative TIPS.

Chang et al. [20] reported matched cohorts of 90 patients (45 for the TIPS group and 45 for no TIPS group) undergoing surgery. The main endpoints were the development of acute-on-chronic liver failure (ACLF), and the 1-year survival. Hernia repair was the most common intervention. The median interval between TIPS placement and surgery was 6 months. This study showed that patients who underwent surgery with a TIPS (median age 63 years) were less likely to develop ACLF at 28 and 90 days, received fewer blood transfusions, had fewer ascites, and had a shorter ICU stay than the control cohort (median age 64 years). Lastly, the TIPS group had a better one-year survival.

Some findings have been obtained in a small study [21] based on patients who underwent low-risk surgery (>80% hernia repairs), with a median of 28 days post-TIPS placement. The control group consisted of patients without TIPS. The 30-day mortality was 0% in both cohorts, and the TIPS cohort developed fewer ascites and acute kidney injury episodes.

A series by Fares et al. [16] reported six cases of hernia repair in patients with refractory ascites. The cohort’s mean age was 61 years. In this study, the median time between TIPS placement and surgery was 24 days. No patient developed severe complications, failure, recurrence, or death after surgery within a mean follow-up period of 2 years.

A retrospective series of Mount Sinai Medical Center [22] showed that, in patients with refractory ascites, complicated by umbilical rupture, preoperative TIPS, followed by a semi-elective repair approach, had a better perioperative and long-term outcome when compared with emergency surgery. The study was designed based on 6 cases of TIPS placement before spontaneous umbilical rupture repair. The median age of patients was 53 years (range: 36–63 years).

## 5. Parenchymal Surgery

None of the wider literature reported this staged approach utilized to facilitate a parenchymal resection; thus, limited definitive recommendations exist regarding optimal perioperative management in this scenario.

Jabbar et al. [23] reported a case of pancreaticoduodenectomy for ampulloma in a patient with congenital liver fibrosis and portal hypertension. TIPS was successfully placed after two attempts. The post-surgical course was complicated by pancreatitis, respiratory distress requiring intensive care unit admission, and TIPS failure diagnosed by CT scan. After conservative management with low molecular weight heparin and broad-spectrum antibiotics, the patient was successfully discharged on postoperative day 28.

A further case of pancreatoduodenectomy, performed 14 days after TIPS placement, was described by Gil et al. [14] in a 60-year-old cirrhotic patient with a CTP score of A6. The indication for pancreatoduodenectomy was pancreatic cancer (TNM stage I, T2, N0). No surgical complications occurred. In the postoperative course, the patients suffered from a slight degree of right cardiac insufficiency due to the excessive increase of the overload and an episode of encephalopathy. After the reduction of TIPS diameter, the patient was admitted to the intensive care unit for three days due to respiratory insufficiency.

Cases of kidney surgery after TIPS placement have been reported.

Azulay et al. [12] described a successful and uncomplicated case of transperitoneal resection for left kidney cancer in a cirrhotic male patient with grade 2 esophageal varices and previous bleeding episodes. The postoperative course was uneventful, but the patient died of cancer recurrence after 24 months.

Schlenker et al. [13] reported a case of nephrectomy in 54-year-old patients with a MELD score of 16, alcohol-related cirrhosis. TIPS was placed to solve refractory ascites 16 days before surgery and the PSPG dropped from 12 to 4 mmHg. Surgery was conducted safely and without postoperative complications. The patient, with a favorable histological diagnosis (stage 1 renal cell carcinoma), was admitted to liver transplantation waiting list after a window of 18 months.

A case of radical nephrectomy for cancer in a patient with advanced hepatopathy CTP score class C cirrhosis was described by Grubel et al. [24]. TIPS was placed 2 months before surgery, achieving a reduction of portosystemic gradient, which dropped from 26 to 14 mmHg.

## 6. TIPS as a Bridge to Liver Transplantation

Liver transplantation is the treatment of choice in patients with terminal hepatic failure secondary to chronic or acute diseases. The main indications are cirrhosis and hepatocellular carcinoma (HCC). Cirrhosis is often complicated by portal hypertension, ascites, esophageal varices, or hydrothorax. TIPS is often employed in the management of these complications when medical therapies fail [34].

Recently, TIPS was involved in the prevention of portal thrombosis, a relative contraindication for liver transplantation.

Several articles aimed to explore the impact of TIPS placement in the setting of liver transplantation [35].

A recent review reported no differences in operative time, transfusion requirement, ICU stay, and length of stay [36].

Regarding long-term outcomes, TIPS placement prolonged transplant-free survival and a reduction of waitlist mortality [37].

No difference was found also in post-transplant patient and graft survival [38,39].

In patients awaiting transplants with a diagnosis of HCC, TIPS placement is considered in order to treat significant portal hypertension and allow bridge percutaneous procedures. Studies investigating the feasibility and safety of radiofrequency ablation (RFTA), radioembolization (TARE), and chemoembolization demonstrated that, in patients with compensated liver function, the results are similar to patients without TIPS [27].

Shunt occlusion or TIPS misplacement represents one of the most frequent complicated conditions during transplant. Nevertheless, as reported by Barbier et al. [40], these complications did not impact the perioperative complications rate.

## 7. Discussion and Conclusions

Despite this review highlighting the incomplete and inconsistent evidence in this field, TIPS as a bridge to abdominal surgery appears feasible and does not jeopardize the ability to perform surgical procedures.

As shown in the current evidence, several types of surgical procedures were performed after TIPS placement, and no differences in postoperative outcomes were found.

It therefore appears evident that the type of procedure should not be considered an absolute contraindication in decompensated cirrhotic patients, even in cases of emergency surgery.

The main advantage of the TIPS placement seems to be the reduction of postoperative ascites occurrence rate. No significant beneficial effects in terms of blood loss, transfusions, post-surgical complications, ACLF, and mortality are shown [41].

Moreover, the postoperative outcome influences the type of surgery. Future randomized prospective trials must stratify the benefit of TIPS placement, based on different types of surgical procedures.

Heterogeneity also involves patient selection, the severity of the liver disease and portal hypertension, the indication for TIPS, and the timing of TIPS placement before surgery [42].

Due to the complexity of procedures, the management of these high-risk patients should be restricted to high-volume centers by surgeons, anesthesiologists, interventional radiologists, and hepatologists with proven experience.

Careful management of the patients before planned surgery, including optimization of pulmonary and cardiac function, the administration of antibiotics and fresh frozen plasma, and the use of diuretics or paracentesis to minimize ascites is recommended in order to avoid major complications [13].

Wide discordance is reported in the literature about the timing between TIPS positioning and surgery. Fares et al. [16] reported a median interval of 25 days after TIPS while Gil et al. [14] were between 14 and 45 days. There are several factors that need to be considered, including the type of surgery, the indication for neoadjuvant therapies, the local expertise, the availability of TIPS, and the resolution of ascites and varices. Moreover, after the procedure of placement, the patient could suffer from transient deterioration of liver function and systemic hemodynamic adaptation to the procedure. The delay could influence the final outcome, especially in cancer patients due to the potential negative impact on oncological results of delayed time to surgery, and in case of complicated benign diseases such as umbilical hernia with life-threatening skin breakdown.

More evidence is needed to optimize the timing of TIPS placement before surgery within a “therapeutic window” in order to maximize the beneficial effects on postoperative morbidity.

It is important to highlight that the category of patients who can benefit from preoperative TIPS are patients with mild to moderate underlying hepatic dysfunction (CTP A or B) and cirrhosis complications, namely refractory ascites and extensive abdominal varices, or both. These patients could experience a high postoperative complications rate, but less likely post-surgery hepatic failure [43].

Patients with CTP class B or C could also benefit from TIPS placement with a gain in terms of survival benefit, but more studies are needed to support this hypothesis [7].

Calls for urgent prospective and well-designed studies that stratify for the preoperative stage are mandatory in this setting.

Finally, the decision for preoperative TIPS placement in cirrhotic patients planned for abdominal surgery needs individualization in the context of multidisciplinary decisions with radiologists, hepatologists, anesthesiologists, and surgeons.

## Figures and Tables

**Figure 1 jcm-13-02213-f001:**
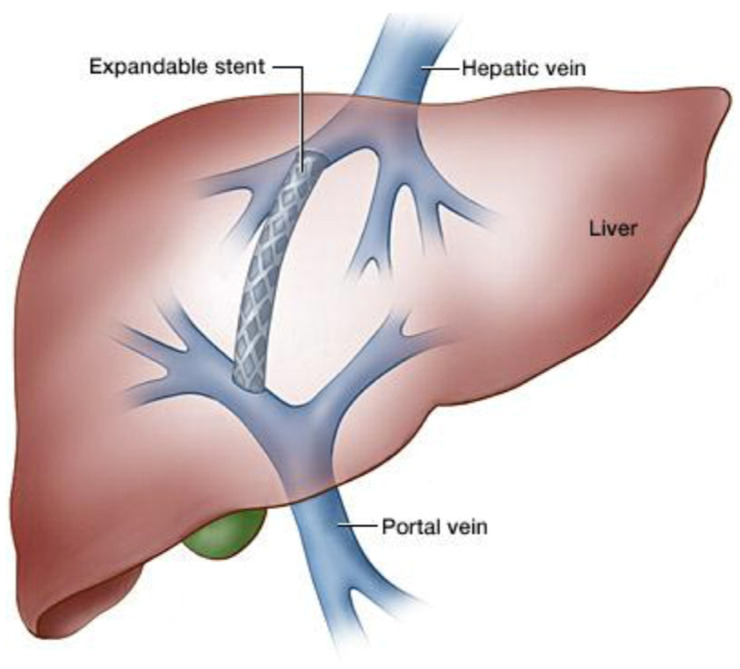
TIPS procedure.

**Table 1 jcm-13-02213-t001:** Study reporting TIPS placement before abdominal surgery.

Study	Stage of Cirrhosis	Number of Cases	Surgical Procedure	Time between TIPS Placement and Surgery	Major Complications
Colorectal surgery					
Vinet, Canada, 2006 [9]	NR	10 TIPS group NO TIPS group	Colon resection	NR	NR
Tabchouri, France, 2019 [4]	CTP A,B,C	30 TIPS group 38 NO TIPS group	Colon resection	Median 40 days	No perioperative major complications (90-day mortality: 1 in TIPS group, 4 in NO TIPS group)
Goel, UK, 2020 [10]	CTP A	15	Colectomy	38 days	NR
Kepeleris, UK, 2022 [11]	CTP A:1 CTP B: 1	2	Colon resection	NR	NR
Azulay, France, 2001 [12]	CTP A and B	2	Colon resection	NR	Death of one patient
Schlenker, US, 2009 [13]	CTP A and B	1	Colon resection	Median 13 days	Ascites; wound infection treated with drainage and antibiotics
Gil, Spain, 2004 [14]	CTP B7	1	Right hemicolectomy	30 days	Right cardiac insufficiency, encephalopathy
Masood, US, 2020 [15]	NR	1	Laparoscopic right hemicolectomy	60 days	No perioperative major complications
Fares, France, 2018 [16]	CTP A-B	6	Colon resection	NR	Ascites and wound infection
Upper-Gastrointestinal Surgery					
Schmitz, US, 2020 [17]	CTP A	8	Sleeve gastrectomy (6), gastrectomy (1), esophagectomy (1)	Mean 38, 7 days	No perioperative major complications
Vinet, Canada, 2006 [9]	NR	5 TIPS group 1 NO TIPS group	Gastrectomy	NR	NR
Schlenker, US, 2009 [13]	CTP A	1	Gastrectomy	Mean 13 days	No perioperative major complications
Gil, Spain, 2004 [14]	CTP A5	1	Subtotal gastrectomy	45 days	No perioperative major complications
de Andres, Spain, 2020 [18]	CTP class A	1	Laparoscopic Heller myotomy + dor fundoplication	42 days	No perioperative major complications
Becq, France, 2015 [19]	NR	1	Antrectomy	90 days	No perioperative major complications
Fares, France, 2018 [16]	CTP A-B	3	Gastric or duodenum resection	NR	NR
Wall surgery					
Chang, Germany 2022 [20]	CTP A or B	11	Hernia repair	Mean 6 months	Ascites, HE, ACLF, infections, blood transfusions
Aryan, UK, 2022 [21]	NR	21 TIPS group 13 NO TIPS group	Hernia repair	median 8 days	Rate of ascites, HE, infections, AKI higher in NO TIPS group
Fares, France, 2018 [16]	CTP B-C	5	Hernia repair	Median 24 days	No major complications
Telem, US 2010 [22]	NR	6	Hernia repair	One day	No major complications
Parenchymal Surgery					
Vinet, Canada, 2006 [9]	NR	2 TIPS group 3 NO TIPS group	Pancreatoduodenectomy nephrectomy	NR	Ascites, HE, infections: no significative reduction in TIPS group
Jabbar, UK, 2016 [23]	NR	1	Pancreatoduodenectomy	NR	No major complications
Gil, Spain, 2004 [14]	CTP A 6	1	Pancreatoduodenectomy	14 days	No major complications
Azulay, France, 2001 [12]	CTP A 6	1	Kidney resection	NR	No major complications
Schlenker, US, 2009 [13]	CTP A	1	Nephrectomy	Mean 13 days	No major complications
Grubel, US, 2002 [24]	CTP C	1	Nephrectomy	56 days	No major complications

NR: not reported; UK: United Kingdom, US: United States; CTP: Child–Turcotte–Pugh Score; HE: Hepatic Encephalopathy; ACLF: Acute on Chronic Liver Failure, AKI: Acute Kidney Impairment.

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
