# Peer review of "Transjugular Intrahepatic Portosystemic Shunt as a Bridge to Abdominal Surgery in Cirrhosis"

_jcm, 2024, doi:10.3390/jcm13082213_

Round 1

Reviewer 1 Report

Comments and Suggestions for Authors The authors conduct a detailed review of the articles published regarding the performance of preoperative TIPS in patients with portal hypertension. In my opinion, it is a well-done work, with a detailed description of the role of TIPS in abdominal surgery. However, some aspects should be improved before its publication. 
  • It should be further investigated into some patient characteristics, such as age, and whether the type of surgery was oncological or not (and analyze if the final outcomes were modified according to the surgery).
  • The times between TIPS placement and intervention are highly variable among the reported data. What do the authors attribute this variability to, and could this potentially influence the final outcomes?
  • It would be advisable to delve further into the discussion and establish a hypothesis regarding which patients could benefit the most from TIPS?
  • It would be advisable to mention cases of patients for whom the placement of TIPS allowed for a surgical intervention that was initially considered contraindicated.
  • There is little information on the adverse effects derived from TIPS, reporting the percentage of encephalopathy, etc.

Author Response

Response to Reviewer #1’ s comments

The authors conduct a detailed review of the articles published regarding the performance of preoperative TIPS in patients with portal hypertension. In my opinion, it is a well-done work, with a detailed description of the role of TIPS in abdominal surgery. However, some aspects should be improved before its publication.

We are grateful to the reviewers for taking the time to read our articles and make pertinent comments. Thank you very much.

  • It should be further investigated into some patient characteristics, such as age, and whether the type of surgery was oncological or not (and analyze if the final outcomes were modified according to the surgery).

Thank you for these valuable suggestions. We modified the text  (in red in emended version), adding requested informatios about age of patients, oncological surgery and outcome according to type of surgery, were available. 

  • The times between TIPS placement and intervention are highly variable among the reported data. What do the authors attribute this variability to, and could this potentially influence the final outcomes?

Thank you very much for your invaluable comment: currently, there is no consensus about the optimal timing of surgery after TIPS placement. There are a numbers of factors that need to be considered, including the type of surgery, the indication for neoadjuvant therapies, the local expertise, the availability of TIPS, and the resolution of ascites and varices. The delay could influence the outcome, especially in cancer patients due to potential negative impact on oncological results of delayed time to surgery, and in case of complicated benign diseases such as umbilical hernia with life threatening skin breakdown.

We emended the text in the “Discussion paragraph” with the suggestions.

  • It would be advisable to delve further into the discussion and establish a hypothesis regarding which patients could benefit the most from TIPS?

Thank you for these valuable suggestions. We implemented the text as follows:

“It is important to highlight that category of patients who can benefit of preoperative TIPS are patients with mild to moderate underlying hepatic dysfunction (CTP A or B) and cirrhosis complications, namely refractory ascites, extensive abdominal varices, or both. These patients could experience high postoperative complications rate, but less likely post-surgery hepatic failure. Patients with CTP class B or C could also benefit from TIPS placement with a gain in terms of survival benefit, but more studies are needed to support this hypothesis”.

  • It would be advisable to mention cases of patients for whom the placement of TIPS allowed for a surgical intervention that was initially considered contraindicated.

Thank you for this suggestion. To date, no sufficient data are available about this point. Recent systematic rewiew reported that all but one planned surgery, was performed after TIPS placement

  • There is little information on the adverse effects derived from TIPS, reporting the percentage of encephalopathy, etc.

Thank you for this suggestion. The complications of TIPS placement are reported at page 4, from line 104; We provided additional informations at the end of paragraph.

Reviewer 2 Report

Comments and Suggestions for Authors

A comprehensive review of a pertinent topic

Comments on the Quality of English Language

The English language is largely acceptable 

Author Response

We thank the reviewers

Reviewer 3 Report

Comments and Suggestions for Authors

The paper provides an interesting review on TIPS as an appropriate procedure to prevent surgical complications in the cirrhotic patient with portal hypertension.

The topic is interesting and current.

Several papers are reported and different types of surgery are addressed. The review is comprehensive and adequate although the topic cannot lead to unequivocal conclusions given the great heterogeneity of the studies analyzed in terms of case histories and procedures. In addition, patient’s populations are small and often limited to case reports.

The authors conclude that prospective studies on the topic should be considered and that in any case, individual cases should be discussed at a multidisciplinary level and managed in expert centers.

Table 1 is very long and the formatting is confusing (acronyms need to be standardized, either CTP or Child is used); text in each cell should be concise or at least the distinction between individual cells should be clear.

Figure 1 is illegible because it is too dark; in any case, the indications to TIPS could also be described in the text in favor of a figure that anatomically illustrates the procedure.

Comments on the Quality of English Language

Extensive review of English is required; typos and punctuation errors are also present.

Author Response

The paper provides an interesting review on TIPS as an appropriate procedure to prevent surgical complications in the cirrhotic patient with portal hypertension.

The topic is interesting and current.

Several papers are reported and different types of surgery are addressed. The review is comprehensive and adequate although the topic cannot lead to unequivocal conclusions given the great heterogeneity of the studies analyzed in terms of case histories and procedures. In addition, patient’s populations are small and often limited to case reports.

The authors conclude that prospective studies on the topic should be considered and that in any case, individual cases should be discussed at a multidisciplinary level and managed in expert centers.

We are grateful to the reviewers for taking the time to read our articles and make pertinent comments. Thank you very much.

Table 1 is very long and the formatting is confusing (acronyms need to be standardized, either CTP or Child is used); text in each cell should be concise or at least the distinction between individual cells should be clear.

Thank you for pointing this out, we do apologize for the mistakes. The table has been modified based on your suggestions. Unfortunately, the graphical layout of the table was created by the editorial office following Journal guidelines

Figure 1 is illegible because it is too dark; in any case, the indications to TIPS could also be described in the text in favor of a figure that anatomically illustrates the procedure.

Thank you for these valuable suggestions. We decided to delete the figure in favor of a figure that anatomically illustrates the procedure.

Round 2

Reviewer 1 Report

Comments and Suggestions for Authors

Thanks for the modifications, congratulations on the work.